# Factors That Affect the Usage Intention of Virtual Learning Objects by College Students

Diana Gaviria [1,*], Juan Arango [1], Alejandro Valencia-Arias [2], Lucia Palacios-Moya [3], Rosa Velez Holguin [4] and Ada Lucia Gallegos Ruiz [5]

1    Facultad de Ciencias Económicas y Administrativas, Instituto Tecnológico Metropolitano, Medellín 050034, Colombia
2    Facultad de Ingeniería, Corporación Universitaria Americana, Medellín 050012, Colombia
3    Centro de Investigaciones, Institución Universitaria Escolme, Medellín 050012, Colombia
4    Dirección Académica, Fundación Universitaria Católica del Norte, Santa Rosa de Osos 051860, Colombia
5    Centro de Investigación y Estudios de la Mujer, Universidad Ricardo Palma, Lima 15039, Peru
*    Correspondence: dianagaviria@itm.edu.co

**Abstract:** In recent years, the introduction of Information and Communications Technologies (ICTs) into several aspects of daily life has become more relevant, particularly in higher education. This phenomenon has resulted in a digital integration that involves the infrastructure of higher education institutions as well as the materials used in the teaching process. Therefore, a study was conducted to investigate the factors that determine usage intention of Virtual Learning Objects (VLOs) by college students in finance and accounting programs in the city of Medellín. This quantitative research employed 86 self-administered questionnaires that enabled us to identify a positive association between attitude toward usage, subjective norms, and the intention to employ this type of content as a learning strategy. In conclusion, models like this need to be developed to understand the acceptance of new technologies in learning processes from different perspectives, and even more so considering that most of the existing literature regarding the subject has been focused on qualitative studies.

**Keywords:** quantitative analysis; online learning; student behavior; computer-aided teaching; higher education

## 1. Introduction

During the last decades of the 20th century, the world faced a series of substantial changes associated to two main factors. One of them was related to advances of Information and Communications Technologies (ICTs). The second one was the expansion of a new economic model that transformed even governing styles. This model consolidated a welfare state—also known as Keynesianism in the literature—since the crisis in 1929 by strong market intervention, which was later discredited by neoliberalism in view of the new dynamics of the following century [1,2].

As a result, the last 20 years have presented great challenges to several stakeholders in education systems. They are concerned about the new ways to improve teaching practices as well as providing students with strategies that make them more competent in globalized contexts, something the 21st century has witnessed [3].

In that sense, education becomes one of the social aspects most sensitive to the new dynamics that take place in the world due to phenomena like globalization, recent technological innovations, and the flaws of the capitalist model. This has resulted in dichotomous situations where education is considered a profitable good, subject to capitalization and, at the same time, a service to be provided embracing a more social philosophy [4].

As a result, in recent years, the introduction of ICTs into several aspects of daily life has become more relevant, particularly in education—at the basic as well as professional and university levels. This phenomenon has resulted in a digital integration that involves

not only the infrastructure of the institutions that provide the educational service, but also the materials and resources used in the teaching process. This type of education is justified by the need of the current society for citizens prepared for virtual communities [5,6].

Consequently, the revolution caused by new technologies in society has caused several theoreticians to refer to it as a new age in which human beings have gone from predictable, linear, and homogeneous groups to heterogeneous societies with multiple communication channels and access to information sources. For that reason, technological solutions in different contexts of human life have become popular; furthermore, they are consolidated as necessary actions in interconnected societies, which has produced the diversification of education [7,8].

In this context education mediated by ICTs has been named eLearning. This concept articulates different technological developments with pedagogical actions in learning-teaching processes where the stakeholders—students, faculty, and administrative staff—are physically and temporarily separated. This type of education has gained more popularity due to the increase in Internet access and the reduction of the costs associated with that service [9].

Based on a study by Mesa [10], 1998 could be considered the starting year of online learning in Colombian education. In that year, two universities offered undergraduate programs mediated by digital tools. That author also reveals that, although new technologies were initially incorporated in higher education institutions (HEIs) as a measure to ensure the efficiency of administrative work, and nowadays they are established in their substantive functions.

In this regard, the progress HEIs have made with the incorporation of ICTs in their organizational dynamics of education, research, and extension has been outstanding. Furthermore, the literature has named eLearning or online learning the usage of these new technologies in education, and this hybridization has been acknowledged to be essential to develop skills in several fields of social life, which was confirmed in the 2003 World Summit on the Information Society [11,12].

As a result, universities have become increasingly interested in using ICTs in educational processes, which is due not only to the usefulness of this type of tools, but also to current global dynamics. By definition, eLearning or web-based education is an educational process mediated by the Internet and designed to complement traditional education at universities. It also seeks to reach those students that require distance education because of their lifestyle, location, or particular needs [13].

As a consequence, although ICTs have been progressively introduced into education based on the requirements of the context, this introduction has not been an easy task for HEIs because they must raise professors' awareness to adapt to the changes it involves going from traditional classrooms to virtual environments. This includes the creation of content for these new educational spaces [14].

Contents for environments mediated by new technologies have been called Virtual Learning Objects (VLOs) in the literature. They constitute support material created by multidisciplinary work teams to aid teaching–learning processes and are characterized by being student centered, which is in line with the philosophy of the current education systems. Additionally, they are composed of three elements: content, learning activities, and contextualization [15,16].

Nevertheless, the literature has also discussed important disadvantages of the adoption of VLOs by educational institutions, for instance, the need for previous knowledge and skills, lack of equipment, technological infrastructure and obsolescence, and the digital gap due to aspects such as internet access [17].

In addition, educational institutions do not implement VLOs in the same way because their pedagogy (which should support and guide students and even faculty members), equipment, and communities depend on multiple variables, e.g., the skills of those involved, communication processes, motivation level, other social aspects, and dissimilar

technological needs in different contexts. All these variables determine the effective usage of technology in a diversity of learning environments [18].

For that reason, the design of this type of contents should be articulated with the reality of students' education, which requires considering multiple aspects, such as the pedagogical approach of a specific field of knowledge, the particular characteristics of the population that will interact with them, and the way they learn [19].

Their relevance in educational contexts has led some institutions—e.g., the University of the Amazon in Colombia—to use them to fight problems such as dropping out. Besides, there are many other benefits implicit in this type of content due to the accessibility and reusability [20,21].

Additionally, these types of materials have been used by education systems as a strategy to respond to several challenges, such as learning mathematics for different fields of knowledge, e.g., engineering, accounting, finances, and other disciplines. This has been reflected in different studies that show that virtual environments have mediated the education of the new professionals joining the workforce [22–24].

In the literature, multiple authors have tried to understand the (general or specific) factors involved in students' adoption of and intention to use VLOs. Wang et al. [25] identified the factors that influence the adoption of an e-learning application by IT students in Malaysia. Cabero-Almenara, Fernández-Batanero, and Barroso-Osuna [26] studied the adoption of virtual reality technologies by university students in Seville, Spain. In Latin America, Veytia-Bucheli and Contreras-Cipriano [27] implemented a qualitative methodology to identify the motivational factors that influence the adoption of VLOs for research by students in a master's program in Mexico. Nevertheless, no quantitative study so far has tried to understand the factors that influence the usage of VLOs by university students in Colombia—let alone those in finance and accounting programs.

Consequently, this study was proposed to answer a question: What factors determine the usage intention of Virtual Learning Objects by college students in finance and accounting programs as a strategy to improve their learning process of International Financial Reporting Standards (IFRS)?

## 2. Materials and Methods

Exploratory quantitative research was empirically conducted for 16 weeks to identify the factors determine the usage intention of Virtual Learning Objects (VLOs) by college students in finance and accounting programs as a strategy to improve their learning process of International Financial Reporting Standards (IFRS).

An empirical comparison was focused on the subject "Liabilities in Accounting", more specifically the topic "Basic financial instruments of liabilities", which is part of Section 11 of the IFRS for SMEs. This course comprises 4 face-to-face hours of lectures by the professor plus 8 h of independent work per week. The sessions are guided by two educational standards on related topics: external financing sources of liabilities and internal financing sources of capital. Both of them are part of the contents of the IFRS.

The sample was composed of 86 students enrolled in two HEIs in Medellín. The researchers who participated in the study determined the number of students in the sample considering several characteristics such as curriculum, topics, weekly hours, independent study hours, skills, evaluations, references, and individual educational attainment.

In that sense, the participants were divided into two groups. Group A contained 45 students enrolled in the "Associate program in Costs and Budget Analysis" in one of the HEIs in this study. Group B was composed of 41 students from the other HEI in the study enrolled in the "Public Accounting" program.

However, it should be mentioned that the sample was affected by some students who missed classes. As a result, the authors decided that they finished the process, but the instrument designed for data collection (survey) was not self-administered so that the results were not biased.

Additionally, VLOs were created and supported by video tutorials on basic financial instruments in the "Liabilities" course with contents from the IFRS. These objects were applied in the classroom in order to implement this type of learning strategy to renew teaching processes and develop academic autonomy in students. The study was conducted in accordance with the Declaration of Helsinki, and approved by the Ethics Committee of Institución Universitaria Escolme (protocol code PC201501) on 20 October 2021.

In total, six interactive video tutorials were produced following the same steps: script in a PowerPoint slideshow, design of Interactive Learning Objects (ILOs) in GeoGebra, video recording with Camtasia, and subsequent video editing.

Therefore, to identify the main factors that determine the intention to use VLOs by college students, this study proposes a model inspired by the Theory of Planned Behavior (TPB) formulated by Ajzen [28]. The TPB measures intentions to adopt different types of behaviors based on variables such as Attitude Toward Behavior, Subjective Norm, and perceived Behavioral Control.

Among the latent variables and constructs in the model, the first variable is Attitude Toward Behavior (ATB). According to Yuriev et al. [29], ATB is defined as an individual's favorable or unfavorable evaluation of a behavior—in this case, the usage of VLOs by college students. Additionally, Behavioral Control (BC) refers to the voluntary control that individuals report they have over their own behavior [30]. As observed in the paper by Qi and Ploeger [31], since ATB and BC are included in the TPB model, many studies have tried to understand the correlation existing between these two variables. Hence, this paper proposes the following hypothesis:

**H1.** *Attitude Toward Behavior (ATB) has a significant positive effect on Behavioral Control (BC).*

Subjective Norm (SN) is defined as the external pressure that an individual experiences to perform a certain behavior [29], which can shape expectations or codes of behavior. According to Ubillos, Páez, and Mayordomo [32], SN can be explained based on the latent variable ATB. These two variables (i.e., SN and ATB) show a correlation that can explain, in part, the intention to perform a behavior. Thus, the following hypothesis is proposed:

**H2.** *Attitude Toward Behavior (ATB) has a significant positive effect on Subjective Norm (SN).*

In accordance with the theoretical foundations of the TPB, all the latent variables in the model are correlated with each other. For that reason, it is necessary to measure the influence of SN over variables such as Behavioral Control before directly measuring Behavioral Intention [33]. Therefore, the following hypothesis is proposed:

**H3.** *Subjective Norm (SN) has a significant positive effect on Behavioral Control (BC).*

It is worth mentioning that the main aim of this study is in line with that of the TPB, i.e., to understand the factors that explain the intention to perform a behavior [33]. In particular, this paper examines finance and accounting students' intention to use a VLO. As the three constructs defined above (i.e., ATB, SN, and BC) are correlated with each other, they can be used to measure or evaluate usage intention. Consequently, the following hypotheses are formulated:

**H4.** *Attitude Toward Behavior (ATB) has a significant positive effect on Behavioral Intention (BI).*

**H5.** *Subjective Norm (SN) has a significant positive effect on Behavioral Intention (BI).*

**H6.** *Behavioral Control (BC) has a significant positive effect on Behavioral Intention (BI).*

To make the video interactive Applet Descartes tool (Java language program) was employed. Thanks to the latter, users can pause the video and focus on reading or going over some important information; when they are done, the video can be resumed. Other sections of the material contain interactive activities in which the user or student can solve problems or answer questions.

It should be noted that each script considered the following aspects in Section 11 of the IFRS for SMEs: scope of Sections 11 and 12; accounting policy choice; introduction to Section

11; scope of Section 11; basic financial instruments; initial recognition of financial assets and liabilities; Initial measurement; subsequent measurement; amortized cost and effective interest method; impairment of financial instruments measured at cost or amortized cost; fair value; derecognition of a financial liability; and disclosures.

Finally, regarding the ethical considerations that protect the participation of the students in the study, the authors expressed the academic nature of the research to the respondents by means of an informed consent based on principles of transparency and confidentiality.

The instrument used in this study (i.e., a survey) was based on the theoretical model proposed by Ajzen's [28], which is composed of Attitude Towards Behavior and social factors called Subjective Norm, Behavioral Control, and Behavioral Intention. All the variables were taken from the literature on TPB [34–36] and adapted to the context of the intention to use VLOs. The survey was administered between June and December of 2019 (near the end of the academic term) in parallel at two higher education institutions in Medellín, Colombia. The answers were provided on a six-point Likert scale: (0) do not know/no opinion, (1) strongly disagree, (2) disagree, (3) neither agree nor disagree, (4) agree, (5) and strongly agree. Table 1 lists the items evaluated in this paper (originally presented to the students in Spanish).

**Table 1.** Questions associated with each construct.

| Construct | Item | Statement |
|---|---|---|
| Attitude Towards Behavior (ATB) | ATB1 | I could take more advantage of my independent work for the course if I had access to Virtual Learning Objects (VLOs) with video tutorials. |
| | ATB4 | VLOs supported by video tutorials facilitate my learning. |
| | ATB3 | VLOs supported by video tutorials in the classroom enable me to interact with classmates and professors more dynamically. |
| | ATB4 | VLOs supported with video tutorials motivate and stimulate my class participation. |
| Behavioral Control (BC) | BC1 | The media and continuous changes in the environment influence my decision to use VLOs in my learning processes. |
| | BC2 | In VLOs supported by video tutorials, there is a relationship between theory and its application to solve example problems, exercises, and problem situations, among others. |
| | BC3 | Basic knowledge of Information and Communication Technologies (ICTs) is necessary to interact with VLOs supported by videos. |
| | BC4 | I'd like to be autonomous in my learning pace, aided by the applications that VLOs offer depending on my major. |
| | BC5 | My classmates value the applications they have access to through VLOs as educational tools that are useful in their educational process. |
| Subjective Norm (SN) | SN1 | If my classmates started using VLOs, I would implement them in my activities as well. |
| | SN2 | Professors should use VLOs to a greater extent in their teaching processes. |
| | SN3 | I could take more advantage of my independent work for the course if I had access to VLOs with video tutorials. |
| Behavioral Intention (BI) | BI1 | I can adopt VLOs in my study techniques and education. |
| | BI2 | In the near future, I'd be in favor of using VLOs to improve my educational process. |
| | BI3 | I intend to use VLOs to improve my learning processes. |
| | BI4 | It's easy for me to learn how to incorporate VLOs to study the topics in my courses. |
| | BI5 | Using VLOs enables me to have a more significant learning experience. |

## 3. Results

To examine the factors that determine usage intention of Virtual Learning Objects by students in finance and accounting programs, four constructs mediated by 17 indicators were taken as reference. They are listed in Table 2.

**Table 2.** Indicators associated with each construct.

| Construct | Number of Indicators |
|---|---|
| Attitude Toward Behavior (ATB) | 4 |
| Behavioral Control (BC) | 5 |
| Subjective Norm (SN) | 3 |
| Behavioral Intention (BI) | 5 |

Hence, based on the four constructs and 17 indicators selected for the measurement, a reliability and validity analysis was conducted. The first stage of this analysis was exploring each construct to verify that the proposed factorial model was appropriate. For that reason and in accordance with Schumacker and Lomax [37], the first step is confirming the validity of the latent factors in a measurement model. After such confirmation, the structural equation model is validated by a set of goodness-of-fit subindices.

To verify the internal consistency of each construct, the Cronbach's Alpha coefficients were calculated (Table 3); according to Bagozzi, Yi, and Phillips [38] said coefficients should be greater than 0.7.

**Table 3.** Reliability Index—Cronbach's Alpha.

| Factor | Cronbach's Alpha |
|---|---|
| ATB | 0.756 |
| BC | 0.705 |
| SN | 0.774 |
| BI | 0.792 |

On the other hand, to evaluate the feasibility of conducting a factorial study and determine how optimal is the fit of the data in the model, Kaiser–Meyer–Olkin (KMO) indices were found and Bartlett's sphericity test was used.

Such indices compare observed correlation coefficients with partial correlation coefficients, whose values range between 0 and 1. Besides, KMO measures are considered unacceptable when they are less than 0.50 [39]. On the other hand, Bartlett's sphericity test compares the hypothesis that the correlations matrix is not an identity matrix, which indicates the existence of significant intercorrelations among the variables [40]. Therefore, to ensure an adequate factorial analysis, the *p*-value should be less than the given critical level, which generally is 0.05. This indicates that the null hypothesis is rejected. Table 4 confirms that the criteria are met by the KMO index as well as Bartlett's sphericity test.

**Table 4.** Convergent validation of KMO and Bartlett's sphericity test.

| Factor | KMO Value | Bartlett Value | Meets the Criteria |
|---|---|---|---|
| ATB | 0.741 | 0.00 | Yes |
| BC | 0.694 | 0.00 | Yes |
| SN | 0.675 | 0.00 | Yes |
| BI | 0.813 | 0.00 | Yes |

In this sense, Campbell and Fiske [41] proposed two aspects to evaluate the validity of a construct: convergent and discriminant validity. The first type defines the degree to which multiple tries to measure the same concept coincide. The idea behind it is that two or more measurements of the same construct should highly covary if they are valid measures

of said construct. The discriminant type is the degree to which the measures of different constructs differ. In this case, if two or more concepts are unique, the valid measures of each should not be highly correlated.

To evaluate convergent validity, the standardized factorial loadings were calculated; according to Chin, Gopal, and Salisbury [42], said validity is achieved if the loadings are equal or greater than 0.6, which is observed in Table 5.

**Table 5.** Convergent validity of standardized factorial loadings.

| Construct | Item | Standardized Factor Loadings |
|---|---|---|
| Attitude Toward Behavior (ATB) | ATB1 | 0.72 |
| | ATB2 | 0.76 |
| | ATB3 | 0.79 |
| | ATB4 | 0.78 |
| Behavioral Control (BC) | BC1 | 0.72 |
| | BC2 | 0.66 |
| | BC3 | 0.72 |
| | BC4 | 0.61 |
| | BC5 | 0.71 |
| Subjective Norm (SN) | SN1 | 0.77 |
| | SN2 | 0.87 |
| | SN3 | 0.86 |
| Behavioral Intention (BI) | BI1 | 0.69 |
| | BI2 | 0.85 |
| | BI3 | 0.73 |
| | BI4 | 0.81 |
| | BI5 | 0.70 |

Discriminant validity is achieved if the confidence interval of the correlations between two factors does not contain a value of 1 [43]. Thus, in accordance with the results in Table 6 regarding this concept, as none of the confidence intervals contains 1, discriminant validity is confirmed.

**Table 6.** Confidence Intervals Report.

| | ATB | BC | SN | BI |
|---|---|---|---|---|
| ATB | 1 | | | |
| BC | [0.238], [0.609] | 1 | | |
| SN | [0.276], [0.642] | [0.431], [0.726] | 1 | |
| BI | [0.549], [0.767] | [0.280], [0.636] | [0.222], [0.610] | 1 |

After the reliability and validity analysis, the hypothesis testing was carried out by validating the structural model composed of a set of hypotheses that include cause–effect relationships (single arrow) and correlations (double arrows) among the four established constructs (Figure 1). It should be noted that such validation used a structural equation method.

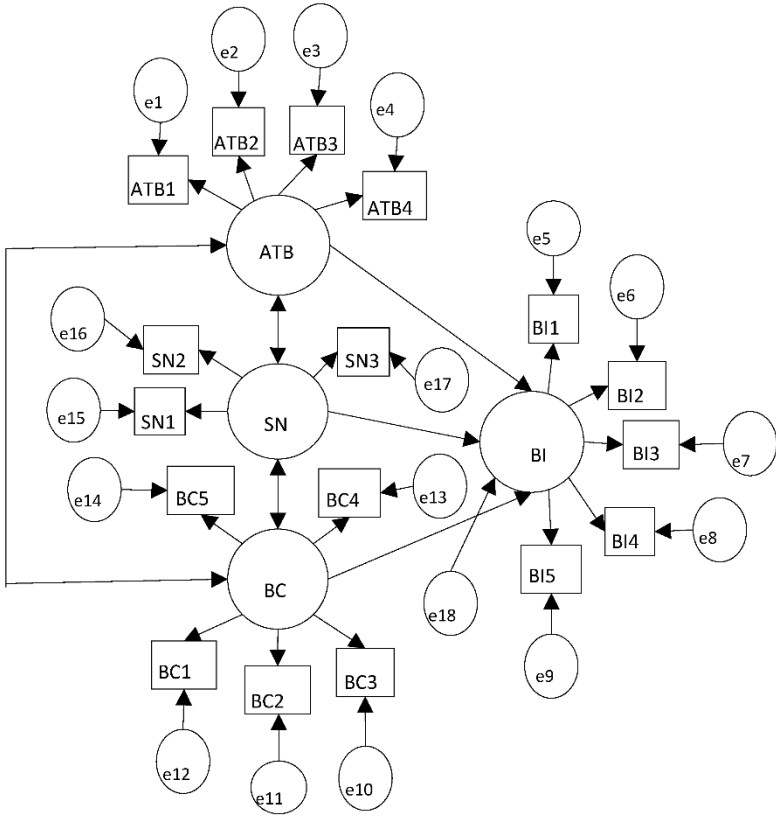

**Figure 1.** Model proposed in the study.

Therefore, to evaluate the fitness of the proposed model with the obtained data, the literature offers some indices that enabled the authors to make such measurement. They are listed in Table 7.

**Table 7.** Goodness-of-fit measurements of the model.

| Index | Measured Value | Threshold | Source | Decision |
|---|---|---|---|---|
| RMSEA | 0.071 | ≤0.06 | Kline (2015) [44] | Does not meet |
| $\chi^2/\mathrm{df}$ | 1.43 | <3 | | Meets |
| CFI | 0.899 | <0.9 | Hair et al. (2006) [45] | Does not meet |
| TLI | 0.878 | <0.9 | | Does not meet |
| SRMR | 0.074 | <0.08 | Hu and Bentler (1999) [46] | Meets |

When the SEM function of the Lavann package [47] was used in software R [48] (THE R CORE TEAM, 2016), the values of indices RMSEA, CFI, and TLI (Table 6) did not reach the threshold. As a result, the model was modified (as shown in Figure 2) by including the following correlations among residuals: BC1 and BC5; SN1 and SN2; BC2 and BC3; and ATB1 and ATB4. They were suggested by the modindices function of said package of R.

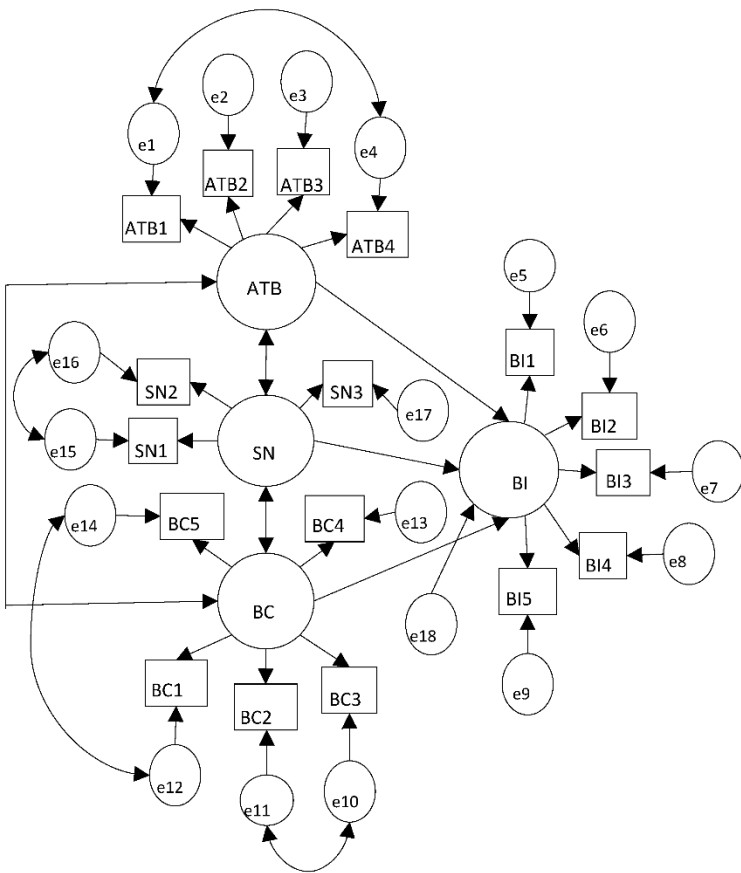

**Figure 2.** Modified model with additional correlations.

The values of the indices in the modified model were the following: RMSEA = 0.06, $\chi^2/df$ = 1.40, CFI = 0.911, TLI = 0.889, and SRMR = 0.074. A better fit was thus achieved because only one index did not reach the threshold. As a result, the hypotheses were tested based on the modified model (Figure 3) using the SEM function of the Lavann package of software R again.

In this sense, Figure 3 provides information regarding the estimations of the beta parameters and *p*-values of each of the six hypotheses in this study. Only one of them, Hypothesis 6, is not supported by the data; in other words, no statistical evidence was found to indicate that perceived Behavioral Control (BC) directly and positively affects Behavioral Intention (BI). By contrast, the remaining cause–effect relationships and correlations are supported. This indicates that Attitude Toward Behavior (ATB) and Subjective Norm (SN) have a positive effect on Behavioral Intention (BI), which is confirmed by the positive sign of the estimation of the beta parameters and the relevance of the corresponding *p*-values.

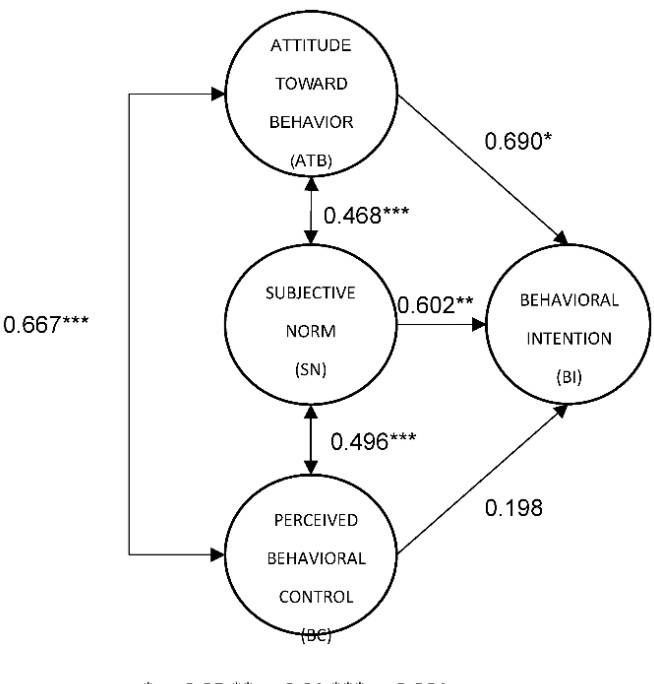

* p<0.05 ** p<0.01 *** p<0.001

**Figure 3.** Research hypothesis testing. * Probability test value less than 0.05, ** Probability test value less than 0.01, *** Probability test value less than 0.001.

## 4. Discussion

According to Chan [49], the current mediation of ICTs in education has been consolidated not only as a strategy to update teaching methods, but also as part of a megatrend of cultural and economic dimensions. Their incorporation has resulted in a paradigm in higher education, where virtual environments are a means of learning–teaching processes, a growing tendency in Latin America [50].

In turn, Rama [51] considers the start of distance education and its different types to be relatively recent there. In that region, since the popularization of the new information technologies—especially during the 1990s—a true reconfiguration of the education systems occurred. New providers of this service have entered the game and many institutions have virtualized their teaching processes. This has enabled the migration to a new conception of student, professor, materials, and classroom in virtual environments.

Consequently, authors such as Torres and Rama [52] acknowledge a series of changes that higher education has faced in the region and the ways education systems are conceived. In this context, virtual environments gain more and more ground every day and are increasingly accepted in several countries in the region, such as Argentina, Ecuador, Panama, Venezuela, and the Dominican Republic.

In this sense, the literature contains several studies that have focused on understanding and analyzing the impact of eLearning on higher education. Some have dealt with materials used as a pedagogical strategy for ICT-mediated environments. This is the case of Virtual Learning Objects, which promote autonomous learning and collaborative work of students and professors, and are reusable, flexible, and enduring [53].

As explained by Quinche and González [54], in the ever-growing field of virtual education, the strategies to motivate students and adapt to the new dynamics of the context are becoming more and more diverse. In this sense, they highlight the use of virtual 3D platforms as new pedagogical strategies to promote innovative and creative actions by key stakeholders in the education process: professors and students. This is in line with the new trend called eLearning or online learning as well as the contents used in it, which are part of the objective of this study.

Therefore, the results were compared to other approaches in the literature, which enabled to confirm not only the importance of this study, but also the need to research this topic further in an ever more globalized society. In our age, education in virtual learning environments is no longer an alternative to include people that due to geographical variables or individual conditions have no access to higher education systems. Instead, it is now an option to diversify the services that have traditionally been offered in face-to-face scenarios, which results in education flexibility [55,56].

One of the most significant findings in this research shows that attitude toward the behavior of using Virtual Learning Objects has a positive influence on usage intention by college students, much more so when they are video tutorials, which enable students to use independent study time more efficiently [57].

In this regard, research conducted by Chalela et al. [58] reveals the importance attached by students of HEIs to online education because it enables them to do curricular and extracurricular activities more efficiently, which matches the results of this study and the findings by Zarceño and Andreu [59]. They highlight the improvement in the use of time by students in digital scenarios. Additionally, regarding the relevance of videos and other materials that ensure a deeper understanding and ease of manipulation of the contents by the students, the case described by Mosquera, Salinas, and Glasserman [60] is a reference that corroborates the results of this study. Their research found the usefulness of multimedia contents for the students of an online guitar course.

Likewise, the work by Jiménez et al. [61] shows the favorable attitude students have to use multimedia materials contained in learning objects for physics courses. Such objects were perceived as a complementary strategy to traditional classroom sessions. Besides, these authors demonstrated the clarity of this kind of pedagogical constructs and their impact on the broadening of knowledge of the use of ICTs.

As a complement to this finding, research by Flores, López, and Rodríguez [62] reveals the main factors students consider when evaluating an online course. For that purpose, they divided the criteria into six categories: technology, design and interface, pedagogy, evaluation, management, and online support. These dimensions compose a measurement model that enabled those authors to complete their task with the participation of 327 students enrolled in different programs at the University of Guadalajara.

In this case, the authors identified three key dimensions that online students pay most attention to when evaluating their satisfaction: (1) A pedagogical element that encompasses the contextual and representational quality of the contents known as Virtual Learning Objects; (2) the interface design that includes the availability of the virtual platform; and (3) online support, which is related to the timely response perceived by student. This is in line with the first finding in this study regarding usage intention of VLO contents by students, much more so when they have an added value often reflected in the access to video tutorials, as mentioned above.

Moreover, another significant result from this study proves that subjective norms—defined as the capacity of the environment to influence an individual—have a positive relationship with students' intention to use Virtual Learning Objects in their education. They are even more relevant if you consider the premise that the subjective norm is a manifestation of social pressure perceived by individuals according to their own thoughts and their peers' take on a particular topic [63,64].

Thus, it can be inferred that the gregarious spirit that characterizes human beings and the permanent seeking for the acceptance from those around them—family, friends, and the society in general—are established as aspects that determine the positive or negative decisions of an individual [65,66].

In the context of this research, the particularities of the subjective norms are key to understand acceptance and the motivation of students to use VLOs in their learning process.

In this context, as pointed out by Caro, Gómez, and García [67], professors play a key role to ensure that students are engaged in the use of new technologies and, as a result,

of the materials provided in online teaching processes. Moreover, teachers are one of the social agents that exert a major influence on students. Said authors studied the public accounting program at Minuto de Dios University in Colombia, and the students expressed the importance of professors' knowledge and usage of ICT tools because, ultimately, they are the ones who share with participants by means of knowledge transfer and generation.

This is in line with the work by Vera and Torres [68], who report the persistence of significant challenges to ensure an adequate adaptation of ICT tools by professors because the construction of new materials congruent with current technological dynamics of education systems precisely depends on such stakeholders. These weaknesses were also mentioned in the Cuban context by Tamayo, Valdés, and Ferras [69] based on their research on the production of VLOs on the island.

However, in addition to the role played by professors in the intention of students to use this type of contents in their learning process, classmates and the general context should also be considered. This is consistent with the ideas of Islas and Delgadillo [70], who stress the influence of the environment over an individual to adopt new technologies for academic processes. Likewise, Barrios [71] states that sociocultural environments and learning processes overlap.

The findings of this study show that the perceived behavioral control of the surveyed students did not influence their behavioral intention. Control implies the perception of internal and external restrictions on a behavior, in this case, adopting VLOs. In line with Kim et al. [72], who studied the acceptance of online learning systems in higher education, perceived behavioral control does not have a positive influence on behavioral intention. This can imply that the expressions of levels of controllability and self-confidence are not so important or fundamental for students when they make decisions regarding the usage and adoption of online learning tools [73].

This result is also supported by the study conducted by Mashroofa et al. [74], who found that behavioral intention is determined by the individual's attitude and subjective norm, but not the perceived behavioral control. Therefore, students perceive that VLO adoption can be hard depending on the available resources, and perceived behavioral control will not increase the usage or adoption of technologies [75].

Based on the above, Rajeh et al. [76] recommend "HEIs to adopt easy-to-use platforms to increase students' confidence to meet their expectations with minimal problems" (p. 6). HEIs should also encourage students' perceived behavioral control and improve their attitudes toward the adoption of VLOs. In turn, professors should support, guide, and train students to improve and ensure their adoption of these learning tools.

Finally, another significant finding of this study was confirming that this type of models—like the one proposed to identify the factors that motivate students to use VLOs in learning processes—are necessary to understand the acceptance of new technologies in learning processes from different points of view [77], and even more so considering that most existing literature regarding this subject has been focused on qualitative studies. Therefore, adopting positivist approaches provides an opportunity to investigate these complex phenomena from other perspectives, which leads to positioning the field of virtual education and the contents produced for such endeavor—Virtual Learning Objects.

## 5. Conclusions

The introduction of new ICTs has become a milestone in modern cultures. These advances have simultaneously led societies to reconfigure themselves in social, political, economic, and cultural terms. As time passes, several daily human activities—e.g., commercial relationships, health service provision, and even education itself—have migrated to new conceptions underlying the consolidation of ICTs.

Moreover, online education has largely been promoted by the popularization of ICTs, and it has enabled countries to apply public policies to expand the reach of different education levels and citizens' access to them. Nevertheless, the challenge that education mediated by these technologies poses for basic, secondary, and professional education

institutions cannot be ignored, because the new paradigm imposed by ICTs leads to reorient educational strategies, including different types of materials designed for that purpose.

In this sense, creating virtual classrooms where the interaction between professors and students can be synchronous or asynchronous—compared to traditional face-to-face education—results in the design of flexible, easy-to-use, reusable materials that are also complex to produce; this is the case of Virtual Learning Objects. Thus, the challenge for institutions—particularly in higher education—is not only the adequate production of contents, but also raising students' awareness so that they know how to work with these objects and enjoy the maximum possible benefit.

Raising awareness is increasingly relevant due to the rapid advance of new technologies in the social, economic, political, and cultural life of current societies. Consequently, the education of new professionals to join the market's workforce demands individuals with advanced digital skills who are able to perform in the context of interconnected communities.

Therefore, promoting the adoption and usage of VLOs by students—regardless of whether their education takes place in totally online environments—is consolidated as a strategy for HEIs and education systems in general to foster more well-rounded education that enables people to be on the same page of current global dynamics. This is especially true in regions like Latin America, where human development and economic growth dynamics have been delayed in comparison with other contexts.

Nevertheless, as revealed by the results and the theoretical framework of this study, ensuring the adoption and usage of Virtual Learning Objects by students is not an easy task because of the influence of several factors, such as the quality of the contents, their interactiveness, and key stakeholders in a student's context—professors and classmates.

Regarding the latter, and especially in the Colombian context, it is logical to ask if professors and students themselves are actually immersed in an education system that has really carried out the necessary process to migrate from a traditional face-to-face scenario to virtual contexts. This new type of environment requires not only the intention to provide the service through new information technologies, but also actual infrastructure that demands additional costs and a paradigm change regarding the implications of said transition. However, any of the stakeholders that influence the decision to use VLOs may experience resistance to change, specifically professors, as mentioned by several authors in the literature.

The findings of the survey administered in this study (in parallel at two HEIs at the end of the semester, during the final exam) indicate that the usage of the VLO by college students was beneficial. Furthermore, the grades of most students in the participating courses were considerably improved with respect to those in other courses that did not apply this kind of learning tools.

Based on these findings, the authors consider relevant to further study this subject to identify the impact of using virtual materials—in addition to the established curriculum—on the education of professionals in the fields of administration, accounting, and finance. Thus, actions can be taken to improve education systems and these professions, because these students will play a key role in the productivity and competitiveness strategies of the territory due to education with a productive, managerial, and strategical approach.

**Author Contributions:** The authors confirm contribution to the paper as follows: conceptualization, D.G. and A.V.-A.; methodology, J.A., A.L.G.R. and A.V.-A.; validation, L.P.-M., A.L.G.R. and D.G.; visualization, R.V.H.; formal analysis, D.G. and A.V.-A.; investigation, J.A.; Project administration, A.L.G.R. and D.G.; software, R.V.H. and D.G.; data curation, A.V.-A. and L.P.-M.; writing—original draft preparation, L.P.-M., R.V.H. and D.G.; writing—review and editing, A.V.-A. and J.A.; supervision, A.V.-A. and A.L.G.R. All authors have read and agreed to the published version of the manuscript.

**Funding:** This work is supported by Instituto Tecnológico Metropolitano, Corporación Universitaria Americana, Fundacion Universitaria Católica del Norte, and Institución Universitaria Escolme, grant number PC201501. This article's article processing charge (APC) were financed by the Instituto

Tecnológico Metropolitano, the Ministerio de Ciencia, Tecnología e Innovación and the Francisco José de Caldas Fund through contract RC282-2022.

**Institutional Review Board Statement:** The study was conducted in accordance with the Declaration of Helsinki and approved by the Ethics Committee of Institución Universitaria Escolme (protocol code PC201501) on 20 October 2021.

**Informed Consent Statement:** Informed consent was obtained from all subjects involved in the study.

**Data Availability Statement:** Data are not publicly available, though the data may be made available on request from the corresponding author.

**Conflicts of Interest:** The authors declare no conflict of interest.

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
