# Peer review of "Factors That Affect the Usage Intention of Virtual Learning Objects by College Students"

_informatics, doi:10.3390/informatics9030065_

Round 1
Reviewer 1 Report
The paper presents a study on the intention to use Virtual Learning Objects (VLOs) by college students in finance and accounting programs in the city of Medellín.
The article is well organized, and the ideas are clearly expressed. The first part of the article describes the importance of the use of ICT and eLearning in education. The use of VLOs potentiates the teaching-learning processes and represents the educational material characterized by being student-centered.
The second part of the article describes the process of creating and using VLOs by college students in finance and accounting programs as a strategy to improve their learning process of International Financial Reporting Standards (IFRS).
The third part of the article explains the results of the study, in table number one the constructors and the number of indicators are listed. The main problem I find is the lack of explanation of the constructors and indicators that are necessary to understand the model that is explained later. In order to understand the model, it would also be interesting to include the questionnaire given to the students, as well as a justification of the questions used from a theoretical point of view.
Next, a validation of the model is carried out, this analysis is very complete and well structured. The problem that I find in the model is that by not having the information on the indicators, their relationships are not understood either. These hypotheses must have a theoretical basis that supports them and that allows the author to understand how the relationships between the four factors.
Finally, in the conclusions, it is missing that they do not comment if after the students have used the VLOs the result in the qualifications has improved. This could be done by comparing the mean grades from other academic courses where VLOs were not used.
Author Response
Medellin, august 23 2022
Dear Antony Bryant
Editor-in-Chief
Informatics
Kind regards
Thank you for the opportunity to revise the paper. We understand the reviewer's comments and we strove to resolve your doubts and concerns. The comments helped us improve the quality of the article and enriched the discussion about it, offering us points of view that we had not considered in the initial version. The following changes were made, properly marked with red letters in the article:
|
Reviewer |
Comment |
Response |
|
Reviewer 1 |
The main problem I find is the lack of explanation of the constructors and indicators that are necessary to understand the model that is explained later. To understand the model, it would also be interesting to include the questionnaire given to the students, as well as a justification of the questions used from a theoretical point of view. |
Theoretical support has been included for the constructs and variables that make up the model to make it more understandable. In addition, the questionnaire given to the students has been included, specifying the theory that serves as a basis for its construction. Lines 157-196 and 212-223. |
|
Reviewer 1 |
Next, a validation of the model is carried out, this analysis is very complete and well structured. The problem that I find in the model is that by not having the information on the indicators, their relationships are not understood either. These hypotheses must have a theoretical basis that supports them and that allows the author to understand how the relationships between the four factors. |
Information on the constructs and variables has been added, justifying by means of theoretical bases and some background information the relationships sought to be tested by means of the hypotheses formulated in lines 157-196. This allows for greater theoretical strength in the proposed model. |
|
Reviewer 1 |
Finally, in the conclusions, it is missing that they do not comment if after the students have used the VLOs the result in the qualifications has improved. This could be done by comparing the mean grades from other academic courses where VLOs were not used. |
The findings have been extended by showing the improvement in grade scores in students who have used the VLOs showing the effectiveness in use. Lines 471-475. |
Thanks in advance for considering the paper for the journal,
Sincerely yours,
The authors

Reviewer 2 Report
I want to thank the authors for providing me the opportunity to review the manuscript titled "Factors that affect the usage intention of Virtual Learning Ob-2 jects by college students". I like that the authors assessed students' intention for Virtual Learning Objects. I have a couple of concerns with the manuscript, which I detail below.
Introduction:
- From the introduction, it can be deduced that the study is about eLearning environments, specifically Virtual Learning Objects (VLO). The authors place the scope of the study in Colombia. However, it is not clear why that scope. The authors can improve the problematization by explaining where the current theories fail to consider the new context and how they hope the insight from this context contributes to theorization.
- Furthermore, the authors address the benefits of VLO but fail to consider the disadvantages, such as the digital divide.
- In addition, different forms of VLO are suitable for different learning activities and environments. Taking into consideration this difference can help to strengthen the contribution of this study.
Review of literature and hypothesis development
- Although the authors mention in the methods section that they developed hypotheses, these hypotheses are not addressed in the paper. Hence, it is impossible to follow the line of argumentation that the authors are testing in this study. One can assume from the methods that the theory of planned behavior was applied, but the authors need to explain how they have applied this theory to study students' intentions in using VLOs.
Methods
- There are a few concerns with the description of the methods used. First, it is not clear how the variables were operationalized. Second, it is also unclear when and how the survey was administered.
Results/ discussion
- The results and discussion seem to be disconnected and can be better aligned. For example, the authors mentioned that one of their findings is that the relationship between behavioral control and intention is not statistically significant. What does it mean for our teaching practices?
Author Response
Medellin, august 23 2022
Dear Antony Bryant
Editor-in-Chief
Informatics
Kind regards
Thank you for the opportunity to revise the paper. We understand the reviewer's comments and we strove to resolve your doubts and concerns. The comments helped us improve the quality of the article and enriched the discussion about it, offering us points of view that we had not considered in the initial version. The following changes were made, properly marked with red letters in the article:
|
Reviewer |
Comment |
Response |
|
Reviewer 2 |
From the introduction, it can be deduced that the study is about eLearning environments, specifically Virtual Learning Objects (VLO). The authors place the scope of the study in Colombia. However, it is not clear why that scope. The authors can improve the problematization by explaining where the current theories fail to consider the new context and how they hope the insight from this context contributes to theorization. |
A paragraph is added in the Introduction acknowledging that there is some research that studies the adoption of OVA in students, however, that these are in other contexts and that in Colombia (and in other countries of emerging economies) these studies are not available in quantitative form, to justify the objective of the research. Lines 113-122. |
|
Reviewer 2 |
Furthermore, the authors address the benefits of VLO but fail to consider the disadvantages, such as the digital divide. |
Information is added mentioning the main disadvantages in the adoption of OVA, supported by the literature in lines 90-99, and mentioning the Reviewer's suggestion regarding the possible disadvantages. |
|
Reviewer 2 |
In addition, different forms of VLO are suitable for different learning activities and environments. Taking into consideration this difference can help to strengthen the contribution of this study. |
Information is added explaining that the adoption of OVA is not uniform, since, understanding the multiplicity of internal and external factors, it varies among different learning environments and social and educational conditions. Lines 113-122 and 157-161. |
|
Reviewer 2 |
Although the authors mention in the methods section that they developed hypotheses, these hypotheses are not addressed in the paper. Hence, it is impossible to follow the line of argumentation that the authors are testing in this study. One can assume from the methods that the theory of planned behavior was applied, but the authors need to explain how they have applied this theory to study students' intentions in using VLOs. |
The theoretical justification of the hypotheses addressed has been made explicit in order to give more clarity to the line of argumentation of the study in lines 157-196. Additionally, more details are given on the application of the Theory of Planned Behavior in the study and its importance in understanding the adoption of OVA. |
|
Reviewer 2 |
(Methods) There are a few concerns with the description of the methods used. First, it is not clear how the variables were operationalized. Second, it is also unclear when and how the survey was administered. |
Information has been expanded in the methodology section on the variables, their origin, types of questions, scales, among other elements that improve the replicability of the study. Lines 212-223. |
|
Reviewer 2 |
(Results / Discussion) The results and discussion seem to be disconnected and can be better aligned. For example, the authors mentioned that one of their findings is that the relationship between behavioral control and intention is not statistically significant. What does it mean for our teaching practices? |
The discussion section has been expanded to provide a better connection with the results in order to better justify the findings obtained in the study and to show their implications when contrasted with the existing literature. Lines 402-419. |
Thanks in advance for considering the paper for the journal,
Sincerely yours,
The authors
